# Neural stem cell temporal patterning and brain tumour growth rely on oxidative phosphorylation

Jelle van den Ameele[1,2], Andrea H Brand[1,2]*

[1]The Gurdon Institute, Cambridge, United Kingdom; [2]Department of Physiology, Development and Neuroscience, University of Cambridge, Cambridge, United Kingdom

**Abstract** Translating advances in cancer research to clinical applications requires better insight into the metabolism of normal cells and tumour cells in vivo. Much effort has focused on understanding how glycolysis and oxidative phosphorylation (OxPhos) support proliferation, while their impact on other aspects of development and tumourigenesis remain largely unexplored. We found that inhibition of OxPhos in neural stem cells (NSCs) or tumours in the *Drosophila* brain not only decreases proliferation, but also affects many different aspects of stem cell behaviour. In NSCs, OxPhos dysfunction leads to a protracted $G_1$/S-phase and results in delayed temporal patterning and reduced neuronal diversity. As a consequence, NSCs fail to undergo terminal differentiation, leading to prolonged neurogenesis into adulthood. Similarly, in brain tumours inhibition of OxPhos slows proliferation and prevents differentiation, resulting in reduced tumour heterogeneity. Thus, in vivo, highly proliferative stem cells and tumour cells require OxPhos for efficient growth and generation of diversity.

DOI: https://doi.org/10.7554/eLife.47887.001

*For correspondence:
a.brand@gurdon.cam.ac.uk

Competing interests: The authors declare that no competing interests exist.

## Introduction

The observation that some cancer cells rely primarily on aerobic glycolysis for energy and biomass production (the Warburg effect) (*Vander Heiden et al., 2009*; *Warburg, 1956*) has often led to the assumption that the other main source of ATP, mitochondrial oxidative phosphorylation (OxPhos), is dispensable. However, it is becoming increasingly clear that many tumours do require mitochondrial activity for energy and biosynthesis and OxPhos is now frequently exploited as a therapeutic target in cancer (*Gui et al., 2016*; *Molina et al., 2018*; *Shi et al., 2019*; *Weinberg and Chandel, 2015*). OxPhos takes place at the inner mitochondrial membrane in five large protein complexes (Complex I-V), which together form the respiratory chain. Complexes I-IV transfer electrons from NADH to $O_2$ and use the released energy to translocate protons from the mitochondrial matrix into the intermembrane space. The resulting electrochemical gradient is then used by Complex V (ATP synthase) to generate ATP from ADP. Apart from the production of ATP, OxPhos is also directly involved in the generation of $NAD^+$, orotate, fumarate and reactive oxygen species (ROS) and thus affects many cellular processes, such as nucleotide synthesis (*Birsoy et al., 2015*; *Sullivan et al., 2018*; *Sullivan et al., 2015*), signalling pathway activity (*Chandel, 2014*) and epigenetic modifications (*Lu and Thompson, 2012*). The Warburg effect has since been interpreted as a normal adaptation to the metabolic requirements of proliferation, both in cancer cells and proliferating stem cells (*Vander Heiden et al., 2009*). High glycolytic flux is thought to be required for a constant supply of biomass while OxPhos, apart from its role in production of ATP, primarily maintains the cellular redox balance (*Birsoy et al., 2015*; *Sullivan et al., 2015*; *Titov et al., 2016*).

However, metabolic flux in cancer cells can be influenced by extrinsic and intrinsic factors such as substrate availability, oncogenic mutations and the tumour's tissue and cell type of origin (*Hu et al., 2013*; *Mayers et al., 2016*; *Vander Heiden and DeBerardinis, 2017*). Brain tumours in particular recapitulate many features of their tissue of origin and grow along a hierarchy reminiscent of normal brain development (*Azzarelli et al., 2018*; *Genovese et al., 2018*; *Lan et al., 2017*; *Lee et al., 2018*; *Tiberi et al., 2014*). An integrated understanding of the interactions between metabolism and cell identity in vivo, during both tumourigenesis and normal development, is therefore crucial to translate advances in cancer research to clinical applications.

Development of the *Drosophila* central nervous system (CNS) has been used extensively as a powerful reductionist model of human brain development and tumourigenesis in vivo (*Brand and Livesey, 2011*; *Hakes and Brand, 2019*; *Villegas, 2019*). The CNS of *Drosophila* develops from rapidly cycling embryonic and larval neural stem cells (NSCs) that generate a wide variety of neurons and glia. Neuronal diversity is achieved primarily by spatial and temporal patterning, which confers specific identities on NSCs and their progeny according to their location and developmental time (*Miyares and Lee, 2019*; *Technau et al., 2006*). Neural stem cells (NSCs) in *Drosophila* and mammals are thought to generate ATP through aerobic glycolysis rather than OxPhos, whereas their neuronal progeny switch to mitochondrial respiration upon differentiation (*Agathocleous et al., 2012*; *Beckervordersandforth et al., 2017*; *Hall et al., 2012*; *Homem et al., 2014*; *Lange et al., 2016*; *Tennessen et al., 2014*; *Tennessen et al., 2011*; *Zheng et al., 2016*). Upregulation of aerobic glycolysis, reminiscent of the Warburg effect, has also been described in a number of *Drosophila* tumour paradigms (*Eichenlaub et al., 2018*; *Wang et al., 2016*; *Wong et al., 2019*). However, the interpretation that mitochondrial respiration is dispensable for normal *Drosophila* NSCs (*Homem et al., 2014*) contrasts with the clear requirement for OxPhos to support cell cycle progression in the *Drosophila* eye disc (*Mandal et al., 2010*; *Mandal et al., 2005*; *Owusu-Ansah et al., 2008*). Here, we investigate whether, and to what extent, *Drosophila* NSCs and brain tumours rely on oxidative phosphorylation.

## Results

### OxPhos is required for brain tumour growth and heterogeneity

We first examined whether OxPhos is required in tumours generated by loss of the transcription factor, Prospero (Pros) (*Caussinus and Gonzalez, 2005*; *Choksi et al., 2006*), in which differentiating daughter-cells revert to a NSC-like fate (*Choksi et al., 2006*) (*Figure 1h*). *pros* tumours are invasive upon transplantation and exhibit genomic instability over time (*Caussinus and Gonzalez, 2005*). We used RNAi to knock down subunits of complex I (NDUFS1) or complex V (ATPsynα) in NSCs and tumour cells with a NSC-specific driver, Worniu-GAL4 (*Albertson et al., 2004*). The complex I RNAi line has been validated previously (*Garcia et al., 2017*; *Hermle et al., 2017*; *Owusu-Ansah et al., 2013*; *Pletcher et al., 2019*); expression of the complex V RNAi in NSCs strongly reduced the levels of ATPsynα (*Figure 1—figure supplement 2a–c*). We also assessed mitochondrial morphology by stimulated emission-depletion (STED) super-resolution microscopy of mitochondria-targeted GFP (*Rizzuto et al., 1995*). Both RNAi lines caused fragmentation of mitochondria (*Figure 1—figure supplement 2d–f*), a known consequence of OxPhos dysfunction in mouse and human cells (*Duvezin-Caubet et al., 2006*).

To our surprise, inhibition of OxPhos through knockdown of mitochondrial complex I or V in *pros* tumours caused a decrease in tumour growth and an overall reduction in brain size (*Figure 1a–c,i*). This result was comparable to the effect observed upon inhibition of glycolysis with an RNAi against aldolase (*Figure 1—figure supplement 1a–c*). This suggests that neither glycolysis nor OxPhos are sufficient to support brain tumour growth in vivo.

Next, we tested the requirement for OxPhos in different types of brain tumours. Constitutive activation of aPKC (aPKC-CAAX) leads to symmetric division of NSCs in the *Drosophila* brain (*Lee et al., 2006*) (*Figure 1h*), whereas loss of *brat* results in dedifferentiation of the progeny of type II NSCs (*Bowman et al., 2008*) (*Figure 1h*). In both aPKC-CAAX and *brat* tumours we found that knockdown of the complex I subunit, NDUFS1, strongly inhibited tumour growth and decreased overall brain size (*Figure 1d–g,i*). This was accompanied by a significant decrease in the mitotic index of tumourigenic NSCs (*Figure 1j*), consistent with mitochondrial metabolism playing a key role in regulating

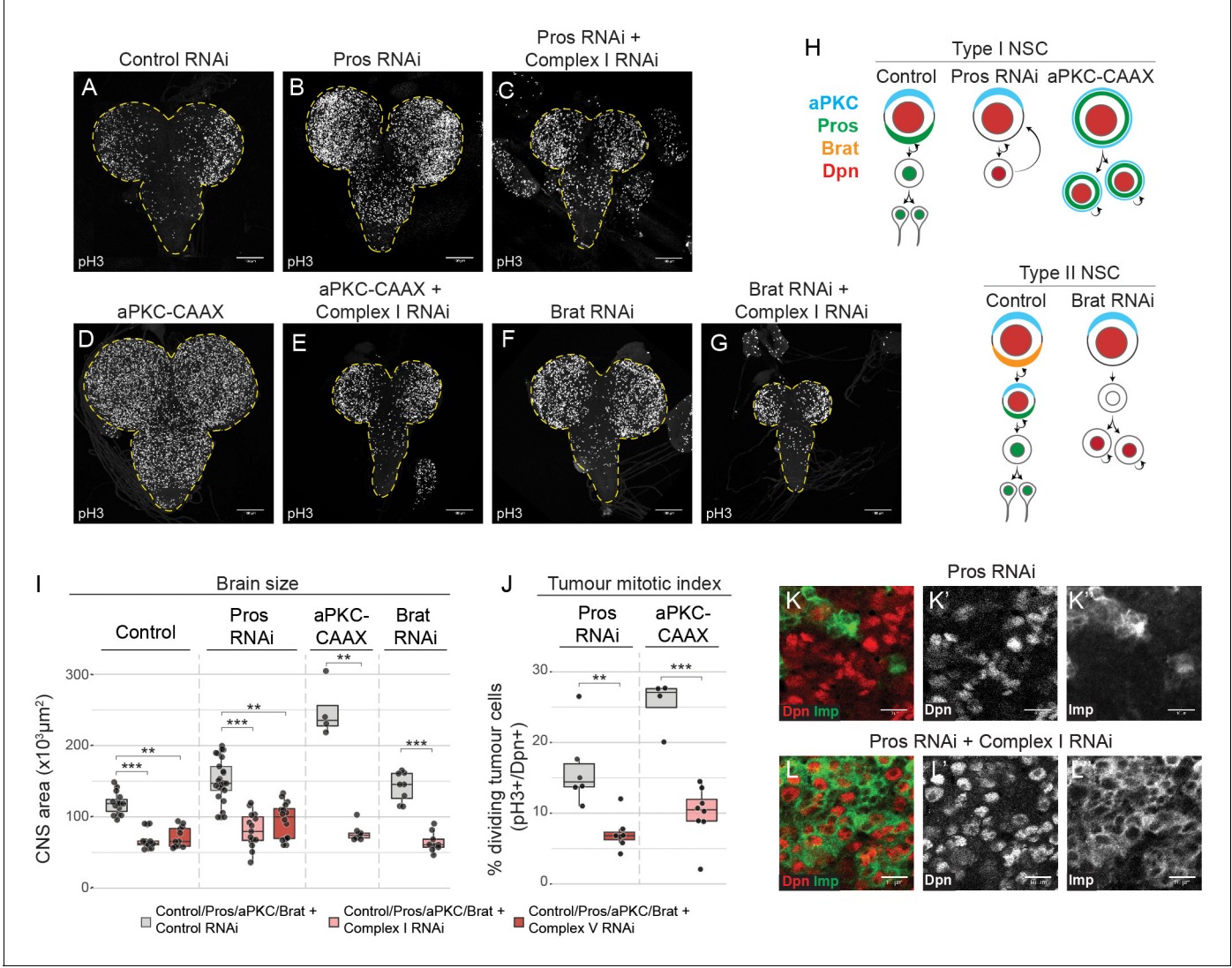

**Figure 1.** Brain tumours require OxPhos for growth. (a–g) phospho Histone H3 (pH3) staining in the CNS of third instar larvae (L3) with NSC-specific expression (Wor-GAL4;Tub-GAL80^ts) of control RNAi (a), Pros-RNAi (b,c), aPKC-CAAX (d,e) or Brat-RNAi (f, g), either without (b,d,f) or with (c,e,g) RNAi against a complex I subunit (NDUFS1). Maximum intensity projections through the entire CNS; dashed lines outline the CNS. (h) NSC lineages before and after tumourigenic transformation. (i,j) Brain size (i) and mitotic index of Dpn+ tumour cells (j) from L3 larvae expressing the indicated transgenes in NSCs. Datapoints indicate individual brains from one to four biological replicates. (k,l) Dpn (red, k',l') and Imp (green, k'',l'') immunostaining in Pros-RNAi tumours, without (k) or with (l) a complex I RNAi. Scale bars are 100 µm (a–g) or 10 µm (k,l).

DOI: https://doi.org/10.7554/eLife.47887.002

The following figure supplements are available for figure 1:

**Figure supplement 1.** Brain tumours require OxPhos and glycolysis for their growth.
DOI: https://doi.org/10.7554/eLife.47887.003

**Figure supplement 2.** OxPhos RNAi in NSCs affects mitochondrial function.
DOI: https://doi.org/10.7554/eLife.47887.004

the proliferation rate of brain tumour cells. There was no obvious increase in apoptosis upon OxPhos inhibition in *pros* tumours, as assessed by TUNEL-staining (*Figure 1—figure supplement 1d–f*).

Growth of *pros* mutant tumours is sustained by a small proportion of highly proliferative stem cells that express Imp (IGF-II mRNA-binding protein) (*Genovese et al., 2018*; *Narbonne-Reveau et al., 2016*). These tumour stem cells self-renew and generate more differentiated Imp-negative tumour cells with limited self-renewal capacity. We assessed whether OxPhos inhibition promotes the differentiation of these Imp-positive stem cells towards Imp-negative tumour cells,

which could result in inhibition of tumour growth (*Genovese et al., 2018*). However, after knockdown of complex I by targeted RNAi, most tumourigenic NSCs in *pros* and aPKC-CAAX tumours remained Imp-positive and differentiation into Imp-negative cells was reduced (*Figure 1k,l—figure supplement 1g*). Our results suggest that OxPhos inhibition does not lead to more aggressive tumours, but rather slows it down by decreasing the proliferation rate of the tumour cells.

## NSC proliferation depends on OxPhos

We found that, as for tumour cells, inhibition of OxPhos in NSCs throughout development resulted in smaller brains (*Figure 1i*; *Figure 2a–d*). This could not be explained by an overall developmental delay, as larval and pupal body length was similar to controls (*Figure 2—figure supplement 1*). In contrast, inhibition of glycolysis by NSC-specific knockdown of phosphofructokinase (PFK), aldolase or phosphoglycerate kinase (PGK) had no effect on brain size and knockdown of pyruvate kinase (PyK) only caused a slight reduction (*Figure 2d—figure supplement 2a-d* and data not shown).

Complex I or V knockdown did not cause an increase in apoptosis in the VNC of third instar larvae (L3) (*Figure 2—figure supplement 2e–k*). However, mitotic index (*Figure 2e*) and incorporation of

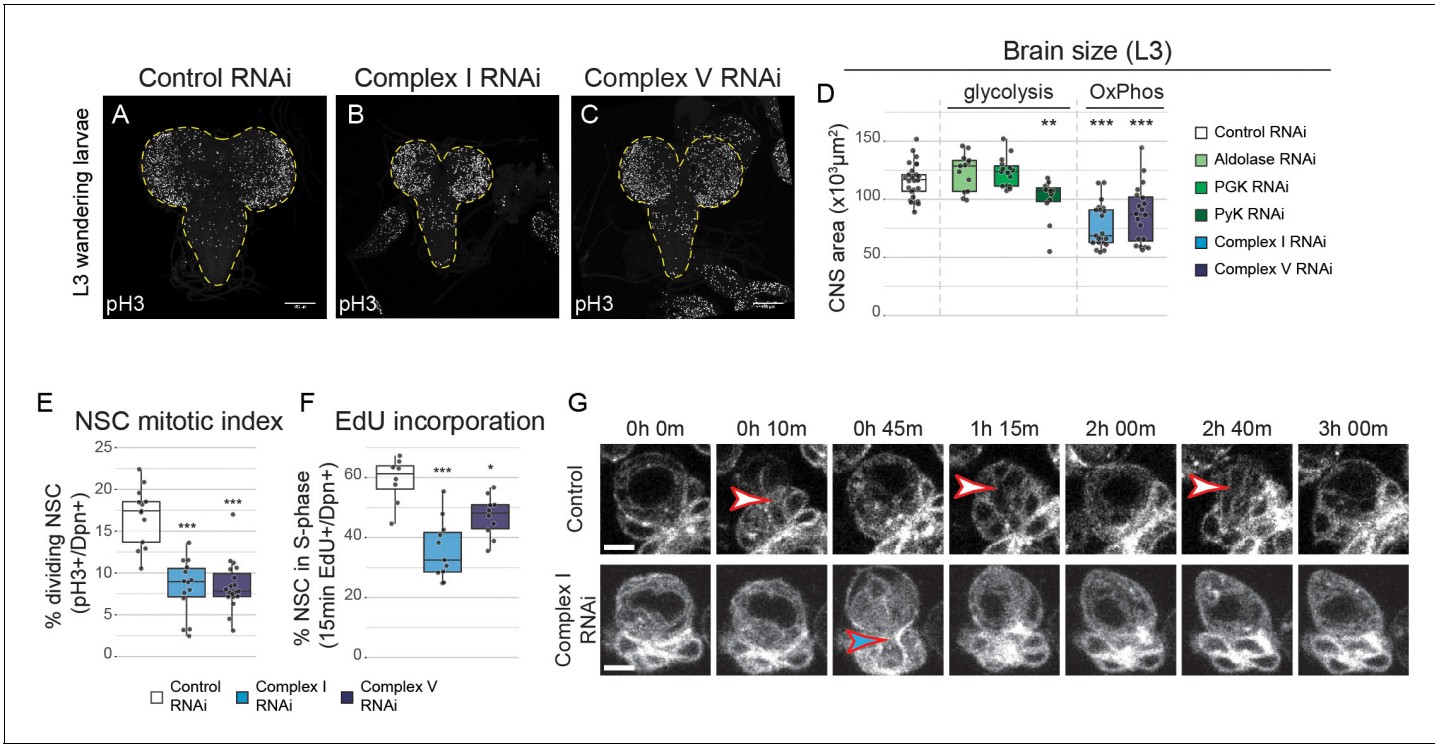

**Figure 2.** OxPhos inhibition decreases NSC proliferation. (a–c) pH3 staining in the CNS of L3 larvae. Maximum intensity projections through the entire CNS; dashed lines outline the CNS. (d) Brain size from L3 larvae. (e,f) Mitotic index (e) and 15 min EdU incorporation (f) in NSCs expressing the indicated RNAi (Wor-GAL4;Tub-GAL80ts). (g) Stills from time-lapse imaging of NSCs (*Figure 2—video 1*) in the early third instar larval VNC with NSC-specific expression of GFP or Complex I RNAi. Arrowheads indicate mitoses of selected NSCs. Datapoints indicate individual brains from four (e), one (f) and two to four (d) biological replicates. Scale bars are 5 μm (g) or 100 μm (a–c).

DOI: https://doi.org/10.7554/eLife.47887.005

The following video and figure supplements are available for figure 2:

**Figure supplement 1.** OxPhos inhibition does not affect body size.
DOI: https://doi.org/10.7554/eLife.47887.006

**Figure supplement 2.** OxPhos inhibition does not increase apoptosis.
DOI: https://doi.org/10.7554/eLife.47887.007

**Figure 2—video 1.** Mitochondrial dysfunction increases cell cycle length.
DOI: https://doi.org/10.7554/eLife.47887.008

**Figure 2—video 2.** Mitochondrial dysfunction increases cell cycle length.
DOI: https://doi.org/10.7554/eLife.47887.009

the S-phase marker 5-ethynyl-2'-deoxyuridine (EdU) (*Figure 2f*) were significantly reduced, indicating that NSCs rely on OxPhos for proliferation. Live imaging of NSCs in the ventral nerve cord (VNC) after complex I knockdown confirmed a striking increase in cell cycle time: NSC division was rarely observed in a 3 hr time window, whereas control NSCs divided between one and three times (*Figure 2g*; *Figure 2—videos 1* and *2*).

To investigate whether RNAi-mediated OxPhos inhibition affects ATP production in NSCs, we measured ATP concentration in vivo using a genetically encoded ATP FRET sensor (*Tsuyama et al., 2013*). ATP concentration in NSCs in the L3 VNC was similar between controls and complex V knockdown (*Figure 1—figure supplement 2g–i*). Acute pharmacological inhibition of glycolysis through application of 2-deoxyglucose to ex vivo cultured brains caused a drop in ATP levels in both conditions. However, this drop was significantly more rapid and severe in NSCs with prior complex V inhibition (*Figure 1—figure supplement 2g–i*). This suggests that mitochondrial dysfunction results in rewiring of NSC metabolism to rely more on glycolysis for ATP production.

## OxPhos is required for temporal patterning of NSCs and their progeny

In order to generate the diversity of neurons and glia within the CNS, NSCs undergo temporal patterning. This allows them to generate progeny with different identities according to their developmental time (*Miyares and Lee, 2019*). *Drosophila* NSCs in the larval VNC progress from an early identity marked by cytoplasmic Imp and nuclear Chinmo, to a late identity marked by cytoplasmic

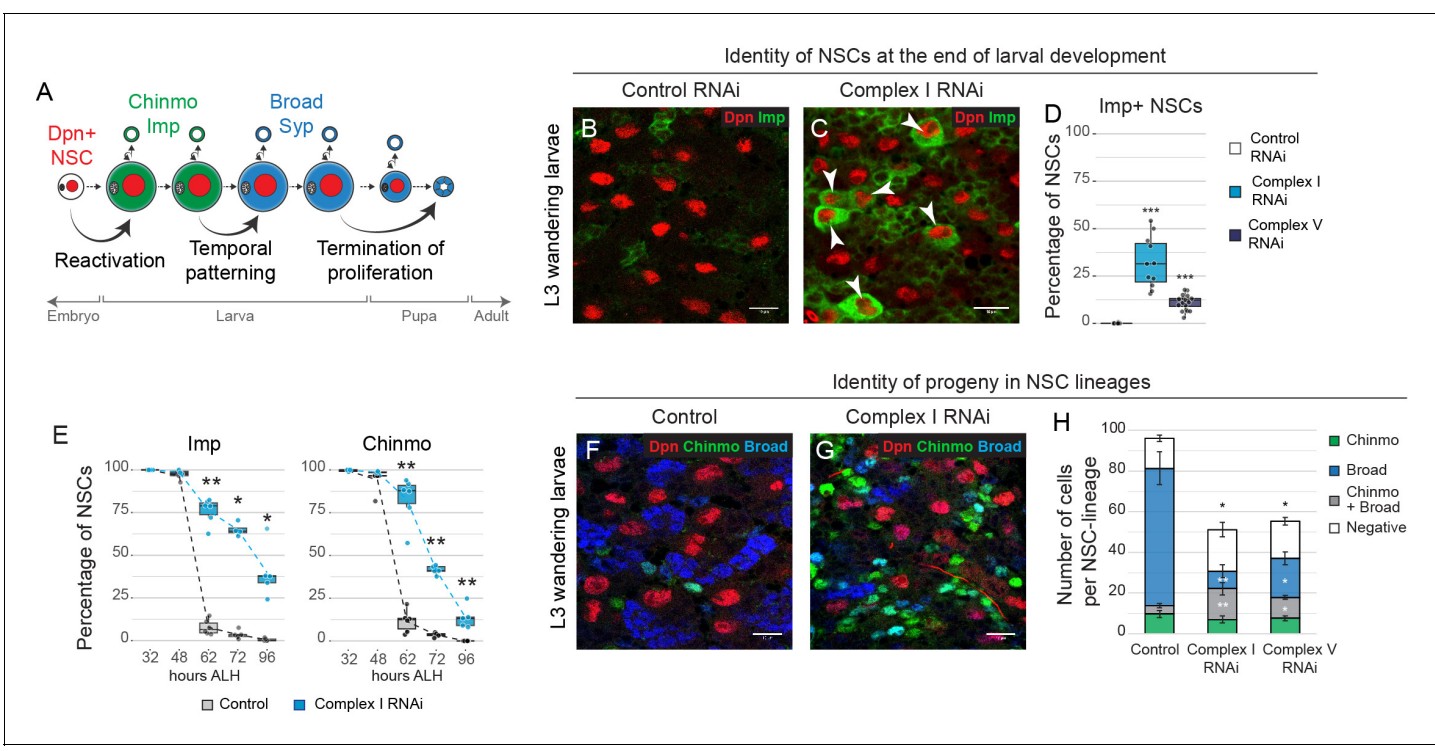

**Figure 3.** OxPhos is required for temporal patterning of NSC and their progeny. (a) Scheme of the major temporal transitions in larval NSCs. (b–d) Dpn and Imp expression in the VNC of L3 larvae. Arrowheads indicate Imp-positive NSCs. (e) Percentage of Dpn-positive NSCs in the thoracic VNC that express the indicated temporal marker at different time points after larval hatching (ALH) at 25°C. (f,g) Dpn (NSCs), Chinmo and Broad in the VNC of L3 larvae. (h) Absolute number of cells per NSC lineage in the VNC that express Chinmo or Broad; graph indicates mean + /- s.e.m. of 6 clones. Datapoints indicate individual brains from four (d), two (e) and one (h) biological replicates. Scale bars are 10 μm.

DOI: https://doi.org/10.7554/eLife.47887.010

The following figure supplements are available for figure 3:

**Figure supplement 1.** Mitochondrial dysfunction in NSCs delays temporal patterning.
DOI: https://doi.org/10.7554/eLife.47887.011
**Figure supplement 2.** Delayed temporal patterning of NSCs affects their progeny.
DOI: https://doi.org/10.7554/eLife.47887.012

Syncrip (Syp) and nuclear Broad (*Liu et al., 2015*; *Maurange et al., 2008*) (*Figure 3a*). We found that inhibition of OxPhos caused a defect in temporal patterning of larval NSCs. After knockdown of complex I, one third of NSCs in the VNC (32.0 ± 4.0%, mean ± s.e.m., n = 11 VNCs) failed to down-regulate Imp expression (*Figure 3b-e—figure supplement 1a-c*) and some (9.1 ± 1.4%, n = 14 VNCs) even failed to differentiate into Syp-positive NSCs at the end of larval life (*Figure 3—figure supplement 1d–g*). This is reminiscent of the failure to downregulate Imp and reduced differentiation in NSC-derived tumours. Immunostaining for other temporal markers (*Maurange et al., 2008*; *Miyares and Lee, 2019*) revealed a delay in the downregulation of the early temporal factors Castor and Chinmo, a decreased peak of expression of the switching factor Sevenup, and delayed upregulation of the late temporal factor Broad (*Figure 3e—figure supplement 1h*). Similar results were observed after knockdown of other subunits of complex I (NDUFA10, NDUFV1) or V (ATPsynα, ATPsynγ) in the VNC (*Figure 3d—figure supplement 1i–m*), and after OxPhos inhibition in the central brain (CB) (*Figure 3—figure supplement 1n–p*). Importantly, this was accompanied by a significant reduction in the number of NSC progeny expressing Broad and lacking Chinmo, indicators of late neuronal identity (*Figure 3f-h—figure supplement 2*). We conclude that OxPhos is required for NSCs to progress from an early to a late temporal fate.

## Temporal patterning of NSCs is regulated at the $G_1$/S transition

To test directly whether increasing cell cycle length inhibits NSC temporal progression, we slowed the cell cycle by expression of Myt1, Wee1 (*Price et al., 2002*) or both, which delay the $G_2$/M transition (*Figure 4g*) and strongly decrease final brain size (*Figure 4—figure supplement 1a*). However, this did not affect NSC temporal progression and no Imp-positive NSCs could be detected at the end of neurogenesis (*Figure 4d–f,h*). Next, we tested whether inhibition of the $G_1$/S transition affects temporal progression by expression of Dacapo (Dap; the p21/p27/p57 homologue), or an activated form of Rb (Rbf280). Strikingly, many NSCs in the VNC expressed Imp continuously (Dap: 7.8 ± 1.2%, n = 15 VNCs; Rb: 24.2 ± 3.4%, n = 10 VNCs) (*Figure 4a–c,h*). When Dap and Rbf280 were co-expressed, a majority of NSCs remained positive for the early NSC marker Chinmo (80.4 ± 1.1%, n = 4 VNCs) (*Figure 4—figure supplement 1b,c*). The block in temporal patterning correlated with the decrease in mitotic index (*Figure 4h—figure supplement 1d,e*). Our data suggest that temporal patterning and generation of neuronal diversity are linked to cell cycle progression and that regulation occurs at the $G_1$/S rather than the $G_2$/M transition.

There is growing evidence for cross talk between mitochondrial metabolism and cell cycle progression at the $G_1$/S transition (*Mandal et al., 2010*; *Mandal et al., 2005*; *Mitra et al., 2009*; *Owusu-Ansah et al., 2008*; *Schieke et al., 2008*). Therefore, we assessed cell cycle stage after knockdown of complex I using Fly-Fucci (*Zielke et al., 2014*) (*Figure 4g*). We found an increase in the number of cells in $G_1$ (26.5 ± 1.6%, n = 9 control VNCs vs. 34.9 ± 1.8%, n = 8 complex I RNAi VNCs) and at the $G_1$/S transition (14.5 ± 2.3%, n = 9 control VNCs vs. 24.3 ± 1.7%, n = 8 complex I RNAi VNCs) (*Figure 4i–k*). Our results suggest that OxPhos dysfunction causes activation of the $G_1$/S checkpoint and this in turn results in delayed temporal patterning of NSCs.

Activation of the $G_1$/S checkpoint upon downregulation of OxPhos activity has been observed in various tissues in *Drosophila* (*DiGregorio et al., 2001*; *Mandal et al., 2005*). In the eye disc, $G_1$/S delay upon complex I dysfunction was caused by increased production of ROS and JNK-pathway activity, while complex IV dysfunction decreased the ATP/AMP ratio and activated the $G_1$/S checkpoint through AMPK and p53 (*Mandal et al., 2010*; *Mandal et al., 2005*; *Owusu-Ansah et al., 2008*). Our preliminary data suggest that decreasing ROS does not rescue the proliferation or temporal patterning defects of complex I or V inhibition (data not shown) and nor does knock down of AMPK or p53 (data not shown). Moreover, clones mutant for *ampk* in a background where all NSCs continue to express complex I or V RNAi enhanced rather than suppressed the temporal patterning defect (*Figure 4—figure supplement 2a–d*). Therefore, it remains to be seen which pathway activates the $G_1$/S checkpoint in NSCs with mitochondrial dysfunction.

## OxPhos dysfunction and prolonged $G_1$/S interfere with termination of proliferation

The adult CNS in *Drosophila* does not normally contain NSCs (*Kato et al., 2009*; *Siegrist et al., 2010*; *von Trotha et al., 2009*). NSCs stop dividing in the first 20–30 hr after pupariation at which

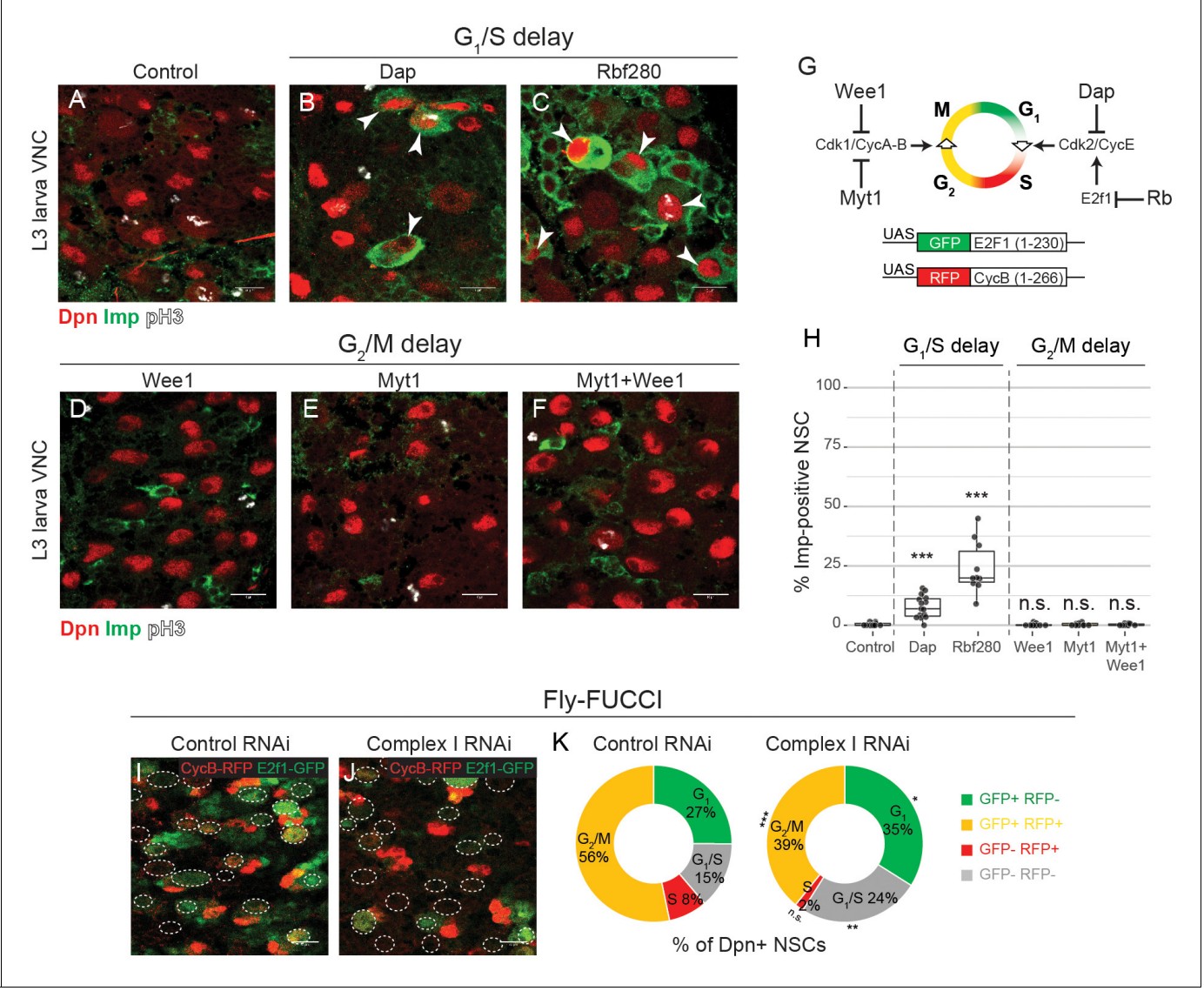

**Figure 4.** $G_1/S$ progression drives temporal patterning. (**a–f**) Dpn (NSCs), pH3 (mitosis) and Imp in the VNC of L3 larvae after NSC-specific expression of the indicated transgene. Arrowheads indicate Imp-positive NSCs. (**g**) Scheme depicting activity of the regulators of the $G_1/S$ and $G_2/M$ transitions that are used for misexpression in this study, and the Fly-FUCCI transgenes. (**h**) Percentage of Dpn-positive NSCs that express Imp in L3 larvae. (**i–k**) L3 larvae with NSC-specific expression of the Fly-FUCCI system, together with control RNAi or complex I RNAi. Outlines indicate Dpn-positive nuclei (**i,j**). The percentage of Dpn-positive NSCs in the VNC that are positive for either GFP ($G_1$), RFP (**S**), a combination of GFP and RFP ($G_2/M$) or none ($G_1/S$ transition); graphs indicate mean of 8 and 9 brains from one biological replicate (**k**). Datapoints indicate individual brains from two or three biological replicates (**h**). Scale bars are 10 μm.

DOI: https://doi.org/10.7554/eLife.47887.013

The following figure supplements are available for figure 4:

**Figure supplement 1.** $G_1/S$ and $G_2/M$ delay results in smaller brains.
DOI: https://doi.org/10.7554/eLife.47887.014

**Figure supplement 2.** AMPK deletion does not rescue the temporal patterning defect caused by OxPhos inhibition.
DOI: https://doi.org/10.7554/eLife.47887.015

time they differentiate or undergo apoptosis (*Figure 3a*) (*Homem et al., 2014*; *Ito and Hotta, 1992*; *Maurange et al., 2008*; *Siegrist et al., 2010*; *Truman and Bate, 1988*). It was previously shown that knocking down complex III or IV subunits in NSCs prevents termination of proliferation at the onset of pupal life (*Homem et al., 2014*). The authors suggested that pupariation is accompanied by a metabolic switch from glycolysis to OxPhos that results in NSC shrinkage and cell cycle

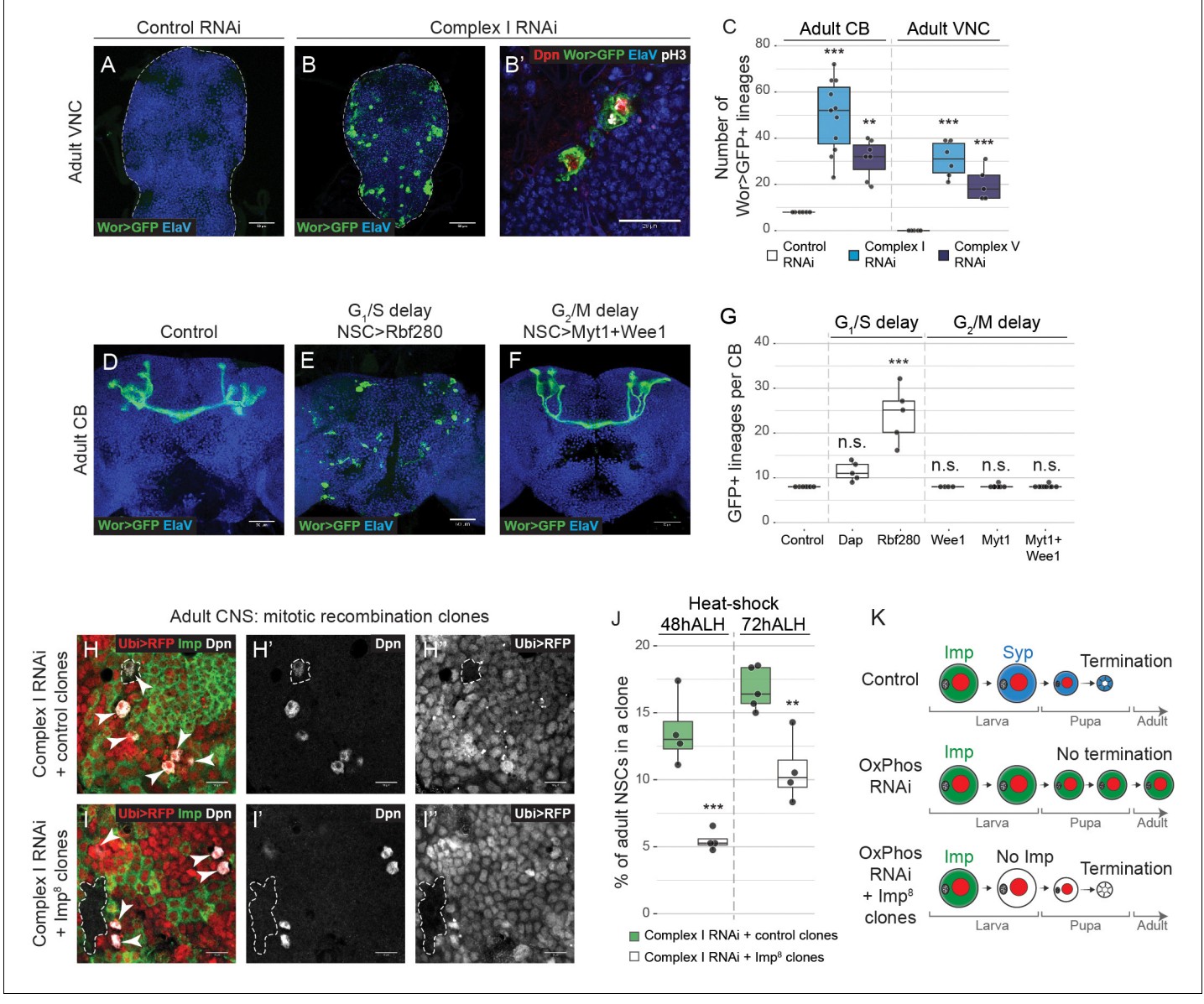

**Figure 5.** NSCs require OxPhos for termination of proliferation. (a–g) ElaV (neurons), GFP (NSCs, Wor-GAL4 >mCD8 GFP), Dpn (NSCs) and pH3 (mitosis) in the pharate adult CB or VNC. Maximum intensity projections through the CB or VNC; dashed lines mark the outline of the CNS. (c,g) Total number of GFP-expressing NSCs in the pharate adult CB or VNC. (h,i) Dpn (NSCs), RFP (negatively marked clones) and Imp in the pharate adult CNS. Arrowheads indicate Dpn-positive NSCs. Dashed outlines mark RFP-negative clones. (j) Percentage of all Dpn-positive NSCs in the pharate adult CNS (CB and VNC) that are part of an RFP-negative clone. (k) OxPhos inhibition prevents terminal differentiation; this is rescued by timely removal of Imp. Datapoints indicate individual brains (c,g) or clones (j) from one biological replicate. Scale bars are 50 μm (a,b), 20 μm (b') or 10 μm (h, i).
DOI: https://doi.org/10.7554/eLife.47887.016

The following figure supplement is available for figure 5:

**Figure supplement 1.** Adult neurogenesis upon OxPhos knockdown and $G_1$/S delay.
DOI: https://doi.org/10.7554/eLife.47887.017

exit. Similarly, we found that when complex I or V subunits were knocked down, NSCs, identified by Dpn-expression and continued expression of GFP from a NSC-specific GAL4-driver (Worniu-GAL4), were maintained into the adult VNC and CB (*Figure 5a-c—figure supplement 1a–d*). Of the 133 NSCs in the larval VNC (*Birkholz et al., 2015*; *Lacin and Truman, 2016*), an average of 30.8 ± 3.1

and $20.2 \pm 3.3$ persisted into adulthood when complex I or V were inhibited respectively (*Figure 5c*). These NSCs continued to proliferate and generate neuronal progeny (*Figure 5a-c—figure supplement 1a-j*). NSCs also persisted in the adult CB and VNC when the $G_1/S$, but not $G_2/M$, transition was delayed, independent of OxPhos dysfunction (*Figure 5d-g—figure supplement 1k-n*).

Timely cell cycle exit of *Drosophila* NSCs at the end of neurogenesis was shown to depend on normal progression through the larval temporal cascade (*Maurange et al., 2008*; *Yang et al., 2017*). We therefore asked whether the defect in termination of proliferation caused by OxPhos inhibition could be due to delayed temporal patterning during larval life, as opposed to a metabolic switch at pupariation. To test this, we restored the temporal identity in NSCs in which complex I was downregulated by removing Imp at 48 hr or 72 hr ALH. Deletion of Imp significantly decreased adult neurogenesis (*Figure 5h–k*), consistent with a direct relationship between temporal patterning defects and the adult persistence of NSCs upon OxPhos dysfunction. Together, our data indicate that the previously observed defect in termination of NSC proliferation is a consequence of the earlier temporal patterning defects caused by OxPhos dysfunction.

## Discussion

Significant progress has been made in identifying the signalling pathways and transcription factors that regulate stem cell transitions during brain development and homeostasis (*Taverna et al., 2014*; *Tiberi et al., 2012*). In contrast, our understanding of the metabolic changes that accompany, or drive, these transitions is still limited (*Knobloch and Jessberger, 2017*). Here we show that the metabolic requirements of highly proliferative NSCs in the *Drosophila* brain, as well as the tumour cells they generate upon transformation, cannot be met by aerobic glycolysis alone. Instead, *Drosophila* NSCs require OxPhos for key aspects of their behaviour: proliferation, generation of diversity through temporal patterning, and termination of proliferation (*Figure 6*). Respiratory activity may provide an explanation for the strong increase in ROS production that has been observed in NSCs upon hypoxia (*Bailey et al., 2015*) and for the developmental lethality caused by CNS-specific mutation of the mitochondrial genome (*Chen et al., 2015*). While OxPhos dysfunction affects both normal NSCs and tumour cells in the brain, inhibition of glycolysis only affects tumour growth (*Figure 1—figure supplement 1*) but not normal brain development (*Figure 2*). This is reminiscent of the upregulation of aerobic glycolysis in Hipk, EGFR or PDGF/VEGF-induced tumours in the *Drosophila* wing disc (*Eichenlaub et al., 2018*; *Wang et al., 2016*; *Wong et al., 2019*). Future experiments will determine the origin and consequences of this tumour-specific reliance on glycolysis in the brain.

Our results contrast with previous findings suggesting that OxPhos is dispensable during normal NSC development and in brain tumours, and is only activated at the end of neurogenesis as part of a metabolic switch to induce termination of NSC proliferation (*Homem et al., 2014*). While our experiments do not directly address whether this metabolic switch takes place, the results provide an alternate interpretation. We find that sustained OxPhos activity throughout NSC development is required for normal temporal patterning. Prolonged expression of early temporal markers makes NSCs unresponsive to the developmental cues that govern cell cycle exit (*Maurange et al., 2008*; *Yang et al., 2017*) and we show that restoring temporal progression by timely depletion of the early temporal factor Imp enhances termination of proliferation in spite of continued OxPhos inhibition. Our findings thus integrate key aspects of NSC and tumour cell biology (*Figure 6*) : OxPhos-dependent proliferation is required for temporal patterning and differentiation at the $G_1/S$ transition of the cell cycle. This enables NSCs to undergo normal aging and to respond to the developmental cues that instruct termination of proliferation. Interestingly, adult neurogenesis in the subventricular zone of the mammalian brain depends on p57-induced slowing of the cell cycle during embryonic development (*Furutachi et al., 2015*). It is not known whether p57 expression or mitochondrial dysfunction also affects the temporal identity of mammalian NSCs. Importantly, the effects we observed are specific to the $G_1/S$ transition: activation of the $G_2/M$ checkpoint did not affect temporal patterning or termination of proliferation. Our results therefore demonstrate that the size and composition of *Drosophila* NSC lineages are not strictly predetermined (*Birkholz et al., 2015*) but rather controlled by both intrinsic and extrinsic factors. Single-cell sequencing data indicate that metabolic differences exist between NSCs in different regions of the brain or at different developmental stages (*Davie et al., 2018*; *Genovese et al., 2018*) and it will be interesting to assess whether all NSCs are similarly affected by OxPhos dysfunction and $G_1/S$ delay or whether specific lineages show

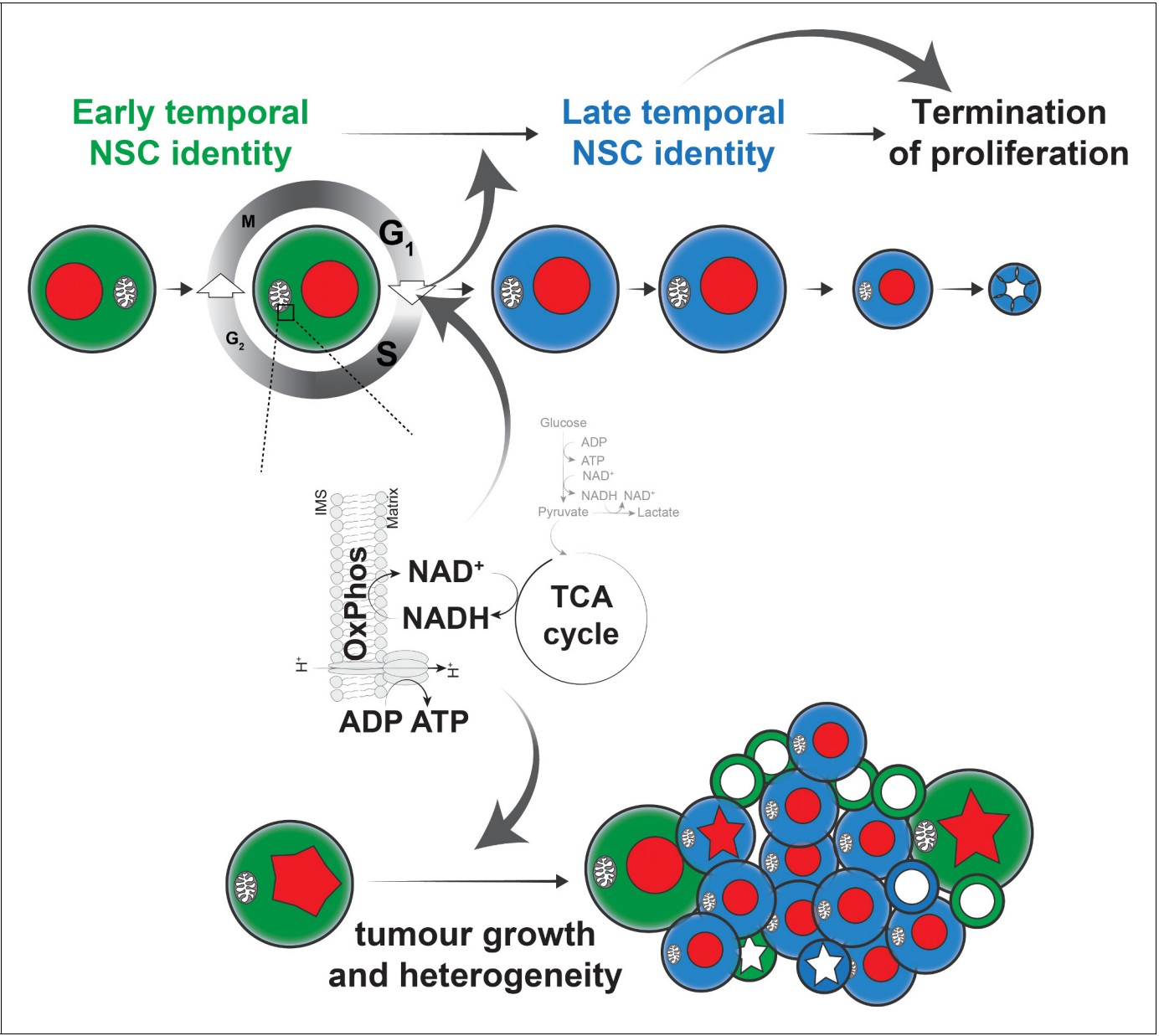

**Figure 6.** Model of the role of OxPhos in Drosophila NSCs and tumour cells. We propose a model, whereby highly proliferative Drosophila NSCs also rely on OxPhos for most aspects of their behaviour. In particular, the $G_1$/S transition depends on OxPhos activity and perturbation of this transition, either directly, or indirectly through OxPhos inhibition, results in delayed temporal patterning. This in turn prevents NSCs from terminating proliferation at the appropriate time, causing neurogenesis to persist into the adult. A similar dependence on OxPhos can be seen in brain tumours, where both proliferation and differentiation require mitochondrial activity, presumably through a similar mechanism to that found in normal NSCs.
DOI: https://doi.org/10.7554/eLife.47887.018

stereotypical responses, as has been shown for entry into quiescence, where arrest in $G_2$ or $G_0$ is predetermined (*Otsuki and Brand, 2018*).

Our study indicates that OxPhos might constitute a targetable metabolic vulnerability of cancer. Small molecule inhibitors of OxPhos are currently being developed and tested in clinical trials to treat various forms of cancer (*Gui et al., 2016*; *Molina et al., 2018*; *Shi et al., 2019*; *Weinberg and Chandel, 2015*). However, we find that the in vivo impact of OxPhos dysfunction is much more

complex than mere inhibition of proliferation. A better understanding of the interactions between metabolism, differentiation and tumour heterogeneity in vivo has the potential to uncover novel therapeutic approaches.

# Materials and methods

**Key resources table**

| Reagent type (species) | Designation | Source or reference | Identifiers | Additional information |
|---|---|---|---|---|
| Genetic reagent (*D. melanogaster*) | mCherry-TRIP | BDSC | RRID: BDSC_35785 | Control RNAi |
| Genetic reagent (*D. melanogaster*) | w[1118];+;+ | BDSC | RRID: BDSC_3605 | |
| Genetic reagent (*D. melanogaster*) | ND75-TRIP | BDSC | RRID: BDSC_33911 | Complex I RNAi |
| Genetic reagent (*D. melanogaster*) | Blw-TRIP | BDSC | RRID: BDSC_28059 | Complex V RNAi |
| Genetic reagent (*D. melanogaster*) | Pros-TRIP | BDSC | RRID: BDSC_42538 | |
| Genetic reagent (*D. melanogaster*) | Brat-TRIP | BDSC | RRID: BDSC_28590 | |
| Genetic reagent (*D. melanogaster*) | ND42-TRIP | BDSC | RRID: BDSC_32998 | |
| Genetic reagent (*D. melanogaster*) | ND51-TRIP | BDSC | RRID: BDSC_36701 | |
| Genetic reagent (*D. melanogaster*) | Blw-RNAi-KK | VDRC | 34663 | |
| Genetic reagent (*D. melanogaster*) | ATPsynβ-TRIP | BDSC | RRID: BDSC_28056 | |
| Genetic reagent (*D. melanogaster*) | ATPsynγ-TRIP | BDSC | RRID: BDSC_28723 | |
| Genetic reagent (*D. melanogaster*) | ATPsynO-TRIP | BDSC | RRID: BDSC_43265 | |
| Genetic reagent (*D. melanogaster*) | PFK-TRIP | BDSC | RRID: BDSC_34336 | |
| Genetic reagent (*D. melanogaster*) | Aldolase-TRIP | BDSC | RRID: BDSC_26301 | |
| Genetic reagent (*D. melanogaster*) | PyK-TRIP | BDSC | RRID: BDSC_35218 | |
| Genetic reagent (*D. melanogaster*) | PGK-RNAi-KK | VDRC | 110081 | |

*Continued on next page*

Continued

| Reagent type (species) | Designation | Source or reference | Identifiers | Additional information |
|---|---|---|---|---|
| Genetic reagent (D. melanogaster) | UASp-EGFP-Myt1 | BDSC | RRID: BDSC_65393 | |
| Genetic reagent (D. melanogaster) | UASt-dWee1 | (*Price et al., 2002*) PMID: 12072468 | | |
| Genetic reagent (D. melanogaster) | UASt-Dap | (*Lane et al., 1996*) PMID: 8980229 | | |
| Genetic reagent (D. melanogaster) | UAS-Rbf-280 | (*Duman-Scheel et al., 2002*) PMID: 12015606 | | |
| Genetic reagent (D. melanogaster) | UASt-aPKC.CAAXWT | (*Lee et al., 2006*; *Sotillos et al., 2004*) PMID: 16357871, 15302858 | | |
| Genetic reagent (D. melanogaster) | UAS-mito-HA-GFP,e1 | BDSC | RRID: BDSC_8443 | |
| Genetic reagent (D. melanogaster) | UAS-AT1.03-NL on III | (*Tsuyama et al., 2013*) PMID: 23875533 | | |
| Genetic reagent (D. melanogaster) | UAS-AT1.03-RK on III | (*Tsuyama et al., 2013*) PMID: 23875533 | | |
| Genetic reagent (D. melanogaster) | UAS-GFP-E2F1.1–230, UAS-mRFP1-NLS-CycB.1–266 | (*Zielke et al., 2014*) PMID: 24726363 | | Fly FUCCI |
| Genetic reagent (D. melanogaster) | Worniu-GAL4 on II | (*Albertson et al., 2004*) PMID: 15536119 | | |
| Genetic reagent (D. melanogaster) | Cas::GFP FlyFos line | VDRC | 318476 | |
| Genetic reagent (D. melanogaster) | Ubi-FRT-Stop-FRT-GFP | BDSC | RRID: BDSC_32251 | |
| Genetic reagent (D. melanogaster) | $Imp^8$ | (*Munro et al., 2006*) PMID: 16476777 | | Imp mutant |
| Genetic reagent (D. melanogaster) | $Ampk\alpha^3$ | (*Haack et al., 2013*) PMID: 24337115 | | AMPK mutant |
| Antibody | rat anti-PH3 (monoclonal) | Abcam | ab10543 RRID: AB_2295065 | IF, 1/500 |
| Antibody | rabbit anti-PH3 (polyclonal) | Merck Millipore | 06–570 RRID: AB_310177 | IF, 1/500 |
| Antibody | guinea pig anti-Dpn (polyclonal) | James Skeath | | IF, 1/10,000 |
| Antibody | rabbit anti-Imp (polyclonal) | (*Geng and Macdonald, 2006*) PMID: 17030623 | | IF, 1/600 |
| Antibody | guinea pig anti-Syp (polyclonal) | (*McDermott et al., 2012*) PMID: 23213441 | | IF, 1/1000 |
| Antibody | chicken anti-GFP (polyclonal) | Abcam | ab13970 RRID: AB_300798 | IF, 1/2000 |

*Continued on next page*

*Continued*

| Reagent type (species) | Designation | Source or reference | Identifiers | Additional information |
|---|---|---|---|---|
| Antibody | rat anti-Mira (polyclonal) | Chris Doe | | IF, 1/500 |
| Antibody | rat anti-Chinmo (polyclonal) | (*Wu et al., 2012*) PMID: 22814608 | | IF, 1/500 |
| Antibody | mouse anti-Broad (monoclonal) | DSHB | 25E9.07 | IF, 1/100 |
| Antibody | rabbit anti-RFP (polyclonal) | Abcam | ab62341 RRID: AB_945213 | IF, 1/500 |
| Antibody | rat anti-ElaV (monoclonal) | DSHB | 7E8A10 | IF, 1/100 |
| Antibody | mouse anti-Sevenup (polyclonal) | (*Kanai et al., 2005*) PMID: 15691762 | | IF, 1/200 |
| Antibody | mouse anti-ATPsynα (monoclonal) | Abcam | ab14748 RRID: AB_301447 | IF, 1/100 |
| Antibody | GFP-booster Atto647N | Chromotek | gba647n RRID: AB_2629215 | IF, 1/500 for STED |
| Commercial assay or kit | ApopTag Red In Situ Apoptosis Detection kit | Merkc Millipore | S7165 | |
| Commercial assay or kit | Click-iT EdU Alexa Fluor 647 Imaging Kit | Invitrogen | C10340 | |
| Chemical compound, drug | 2-deoxyglucose | Sigma | D8375 | 200 mM final concentration |

## Fly husbandry

*Drosophila melanogaster* were reared in cages at 25℃. For most experiments, embryos were collected on food plates for 3 hr and transferred to 29℃ until analysis. Unless indicated otherwise, larvae were matched for developmental timing at wandering third instar (L3). For time-course experiments, embryos were collected on yeasted apple juice plates and larvae were transferred to a fresh yeasted food plate within 2 hr of hatching (designated 0 hr ALH) and grown at 25℃ until the desired stage. For clonal analysis, embryos and larvae were grown at 25℃ and heat-shocked when indicated for 20 min in a 37℃ water bath.

## Fly stocks

The following stocks were used: mCherry-TRIP (Bl#35785) was used as control RNAi and w[1118];+;+ as control. Unless otherwise indicated all Complex I RNAi data are from ND75-TRIP (NDUFS1; Bl#33911)(*Owusu-Ansah et al., 2013*) and all Complex V RNAi data from Blw-TRIP (ATPsynα; Bl#28059)(*Teixeira et al., 2015*). The other UAS-lines used were: Pros-TRIP (Bl#42538); Brat-TRIP (Bl#28590); ND42-TRIP (Bl#32998)(*Garcia et al., 2017*); ND51-TRIP (Bl#36701) (*Garcia et al., 2017*); Blw-RNAi-KK (VDRC#34663) (*Teixeira et al., 2015*); ATPsynγ-TRIP (Bl#28723) (*Teixeira et al., 2015*); PFK-TRIP (Bl#34336); Aldolase-TRIP (Bl#26301); PyK-TRIP (Bl#35218); PGK-RNAi-KK (VDRC#110081); UASp-EGFP-Myt1 (Bl#65393); UASt-dWee1 (*Price et al., 2002*); UASt-Dap (*Lane et al., 1996*); UAS-Rbf-280 (*Duman-Scheel et al., 2002*); UASt-aPKC.CAAXWT (*Lee et al., 2006*; *Sotillos et al., 2004*); UAS-mito-HA-GFP,e1 (Bl#8443); UAS-AT1.03-NL and UAS-AT1.03-RK on III (*Tsuyama et al., 2013*); Fly-FUCCI (*Zielke et al., 2014*) was UAS-GFP-E2F1.1–230, UAS-mRFP1-NLS-CycB.1–266. All RNAis against OxPhos or glycolysis components caused developmental lethality upon ubiquitous expression with Tubulin-GAL4 on III. The GAL4-driver used throughout the study was Worniu-GAL4 on II (*Albertson et al., 2004*), either on its own, or recombined with UAS-mCD8-GFP on II and Tub-GAL80[ts] on III. Castor was visualised with Cas::GFP FlyFos line (VDRC#318476). The genotypes for mitotic recombination clones (*Figure 3h*; *Figure 3—figure supplement 2d–f*; *Figure 4—figure supplement 2*; *Figure 5h–j*) were as follows: yw,FRT19a (control), yw,Imp[8],FRT19a (Imp-mutant;

*Munro et al., 2006*) or yw,Ampkα³,FRT19a (AMPK-mutant; *Haack et al., 2013*)/yw,hsflp,ubi-RFP, FRT19a;Wor-Gal4;+ or ND75-TRIP/+ or Blw-TRIP/+. This resulted in NSC lineages which were randomly marked upon heat-shock by mitotic recombination, whereby all NSCs in the CNS continued to express the RNAi. The genotypes for flip-out clones (*Figure 5—figure supplement 1e–j*) were as follows: yw,hsflp/+;Wor-Gal4;Ubi-FRT-Stop-FRT-GFP (from Bl#32251) (*Evans et al., 2009*)/+ or ND75-TRIP or Blw-TRIP.

## Immunostaining, EdU and TUNEL

Larval brains were dissected in PBS with 0.3% Triton (PBST), fixed in 4% formaldehyde/PBST for 20 min and washed three times in PBST. For EdU incorporation, freshly dissected brains were immersed in PBS containing 200 ug/ml EdU for 15 min, rinsed twice in PBS and then fixed. EdU detection was performed using a Click-iT EdU Alexa Fluor 647 Imaging Kit (Invitrogen C10340) according to the manufacturer's instructions. For immunostaining, brains were incubated with primary antibodies in PBST overnight at 4°C, washed with PBST, incubated with Alexa Fluor-conjugated secondary antibodies (Life Technologies) or a GFP-nanobody coupled to Atto647N (Chromotek gba647n) diluted 1/500 in PBST overnight at 4°C and washed with PBST. Brains were mounted in Prolong Diamond Antifade Mountant (Invitrogen). TUNEL staining was done using the ApopTag Red In Situ Apoptosis Detection Kit (Merck Millipore S7165) according to the manufacturer's instructions.

## Antibodies

The following primary antibodies were used: rat anti-PH3 (1/500, Abcam ab10543); rabbit anti-PH3 (1/500, Merck Millipore, 06–570); guinea pig anti-Dpn (1/10,000, gift of James Skeath); rabbit anti-Imp (1/600, gift of Paul MacDonald; *Geng and Macdonald, 2006*); guinea pig anti-Syp (1/1000, gift of Ilan Davis; *McDermott et al., 2012*); chicken anti-GFP (1/2000, Abcam ab13970), rat anti-Mira (1/500, gift of Chris Doe); rat anti-Chinmo (1/500, gift of Nicholas Sokol; *Wu et al., 2012*); mouse anti-Broad (1/100, DSHB 25E9.07); rabbit anti-RFP (1/500, Abcam ab62341); rat anti-ElaV (1/100, DSHB 7E8A10); mouse anti-Sevenup (1/200, gift of Yasushi Hiromi; *Kanai et al., 2005*); mouse anti-ATPsynα (1/100, Abcam ab14748).

## Imaging and image processing

Fluorescent images were acquired using a Leica SP8 confocal microscope and analysed using ImageJ. For the larval CNS, we imaged the thoracic segments of the VNC from the ventral side until the neuropil, or the ventral regions of the CB; for the adult CNS, the entire VNC or CB was imaged. All images are single sections, unless indicated otherwise. For live imaging, third instar larval brains were dissected at room temperature in Schneider's insect medium (Sigma S0146), mounted in Schneider's medium with 10% FBS on low 35 mm Ibitreat dishes (Ibidi 80136) and imaged on an inverted Leica SP8 confocal microscope at room temperature. Z-stacks of the ventral side of the thoracic VNC were made at the indicated intervals for 3 hr. For live in vivo ATP measurements with an ATP FRET sensor for *Drosophila* (*Imamura et al., 2009*; *Tsuyama et al., 2013*), confocal settings were as follows: 405 nm excitation and simultaneous detection at 445–490 nm (CFP) and 530–760 nm (FRET); 2-DG (Sigma D8375) was added to the medium to a final concentration of 200 mM. Ratios were calculated for mean FRET/CFP intensity per NSC. Stimulated emission depletion (STED) super-resolution imaging was performed on a custom STED microscope as described in *Trovisco et al. (2016)* with a 100x UPlanSApo 1.35 NA silicone oil immersion objective lens (Olympus, Japan) over a region of 20 μm² (1024 × 1024 pixels). Images were processed using ImageJ. Timestamps were generated with a custom-built OverTime ImageJ plugin (Richard Butler). Figures were compiled in Adobe Illustrator.

## Quantifications and statistical analysis

For quantification of NSCs, Dpn- or Mira-positive NSC on the ventral side of the thoracic VNC at the indicated stage were counted. For TUNEL-staining all TUNEL-positive cells were quantified throughout the entire thickness of the thoracic VNC. To quantify adult NSCs, all GFP-positive lineages were counted throughout the entire VNC or CB; in the control CB, GFP perdures until pharate adult stage in eight mushroom body lineages, which terminate proliferation only at the end of pupal life. To quantify adult NSCs upon Imp-mutation (*Figure 5j*), all Dpn-positive cells were counted in the VNC

or CB. Mitotic index is the number of pH3-positive cells among Dpn-positive cells. For quantification of tumour mitotic index, over 200 Dpn-positive cells were quantified in each thoracic VNC. For brain size, the area of CNS maximum projections was measured.

Graphs were generated in R or Excel. Box-and-whisker plots depict median, interquartile range (box) and 1.5IQR below and above the first and third quartiles respectively (whiskers). Bar graphs, line graphs and values in the text indicate mean ± s.e.m. Datapoints indicate the value of individual VNCs or CBs, apart from *Figure 5—figure supplement 1j* where datapoints depict individual clones. One biological replicate is defined as the result of one parental cross.

Statistical tests were performed in R. All datasets were first checked for normal distribution with a Shapiro-Wilk test, and then ANOVA was performed with a post-hoc Tukey test. When data were not normal distributed, Kruskal-Wallis test was performed with post-hoc Dunn test and Bonferroni adjustment for multiple comparisons. For time course experiments (*Figure 3e—figure supplement 1h*), the two conditions at individual time-points were compared with a two-sided Mann-Whitney U test. ATP measurements (AT1.03-NL) were normalised for each NSC to t = 0 when 2-DG was added to the medium, and to the mean values from VNCs that expressed an ATP-insensitive sensor (AT1.03-RK) and were imaged in the same experiment. Modelling of the dynamics of ATP levels was done in R, based on the assumption of exponential decay. Significance is shown compared to control samples, unless indicated otherwise, with the following symbols: $*p<0.05$; $**p<0.01$; $***p<0.001$; n.s. $p\geq0.05$.

## Acknowledgements

We thank C Doe, I Davis, Y Hiromi, P MacDonald, J Skeath, N Sokol for antisera, and D St Johnston, T Uemura, the Bloomington Drosophila Stock Centre and Vienna Drosophila Resource Centre for Drosophila stocks. We thank R Butler (Gurdon Institute Imaging Facility) for the OverTime Fiji plugin, G Sirinakis for help with super-resolution imaging, R Krautz for statistical modelling and Y Bellaiche, W Staels, F Munoz-Martinez, PF Chinnery and members of the Brand lab for discussion and comments on the manuscript. This work was funded by Wellcome Trust Senior Investigator Award (103792) and Royal Society Darwin Trust Research Professorship to AHB. JvdA was supported by EMBO Long-term Fellowship (ALTF 1600_2014) and Wellcome Trust Postdoctoral Training Fellowship for Clinicians (105839). AHB acknowledges core funding to the Gurdon Institute from the Wellcome Trust (092096) and CRUK (C6946/A14492).

## Additional information

### Funding

| Funder | Grant reference number | Author |
| --- | --- | --- |
| Wellcome Trust | 103792 | Andrea H Brand |
| Royal Society | | Andrea H Brand |
| European Molecular Biology Organization | ALTF 1600_2014 | Jelle van den Ameele |
| Wellcome Trust | 105839 | Jelle van den Ameele |
| Wellcome Trust | 092096 | Andrea H Brand |
| Cancer Research UK | C6946/A14492 | Andrea H Brand |

The funders had no role in study design, data collection and interpretation, or the decision to submit the work for publication.

### Author contributions

Jelle van den Ameele, Conceptualization, Resources, Formal analysis, Funding acquisition, Validation, Investigation, Methodology, Writing—original draft, Writing—review and editing; Andrea H Brand, Conceptualization, Resources, Formal analysis, Supervision, Funding acquisition, Writing—original draft, Project administration, Writing—review and editing

## Author ORCIDs

Jelle van den Ameele (iD) http://orcid.org/0000-0002-2744-0810
Andrea H Brand (iD) https://orcid.org/0000-0002-2089-6954

## Decision letter and Author response

Decision letter https://doi.org/10.7554/eLife.47887.021
Author response https://doi.org/10.7554/eLife.47887.022

## Additional files

### Supplementary files

• Transparent reporting form
DOI: https://doi.org/10.7554/eLife.47887.019

### Data availability

All data generated or analysed during this study are included in the manuscript and supporting files.

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
