## [Decision Letter]

Thank you for submitting your article "OxPhos-dependent stem cell proliferation drives temporal patterning and tumour growth in the developing brain" for consideration by *eLife*. Your article has been reviewed by four peer reviewers, and the evaluation has been overseen by Claude Desplan as Reviewing Editor and Utpal Banerjee as the Senior Editor. The reviewers have opted to remain anonymous.

The reviewers have discussed the reviews with one another and the Reviewing Editor has drafted this decision to help you prepare a revised submission. The four reviewers agree on the significance and importance of the paper and the quality of the data presented, especially since the Warburg effect is the topic of so much discussion. It was thought that normal NSC and brat tumors do well without OxPhos while the manuscript proves that this might not be the case. The paper clearly makes this point, which is important.

However, there are several serious criticisms that need to be addressed before the paper can be considered.

- A better presentation of the problem.

- One major concern is that the paper does not really examine glycolysis; it demonstrates that knocking down two essential components of OxPhos compromises normal NSC development as well as larval brain tumor growth. Refocusing the text on OxPhos while discussing the results in the context of Homem et al. and Mandal et al. would significantly improve the manuscript. It should be noted that the manuscript neither confirms nor discards the previously claimed Glycolysis-to-OxPhos transition.

- The lack of metabolic characterization of mutants. A revised version should examine mitochondria morphology, their membrane potential, and ATP levels.

- One reviewer would also like the paper to address the possibility that these are glial effects.

- Delete the over-interpretations. In particular, the title must be changed to reflect this: "drive" is an overstatement as the manuscript shows it to be "necessary".

The reviews below are quite detailed and should help you improve the manuscript for publication.

Reviewer #1:

This manuscript by van den Ameele and Brand addresses the interesting question of metabolic reprogramming during tumor formation or stem cell behavior. Authors use *Drosophila* neural stem cells as a model and show that reducing OxPhos by the knock down of specific components of the respiratory chain complexes, either during uncontrolled NSC proliferation or during normal NSC life, alters their proliferation, blocks the cells in G_1_/S and affects the terminal differentiation of NSCs. This finding may be of interest in the context of many studies showing that tumor cells as well as stem cells require OxPhos for their proliferative activity.

However, the conclusions of the experiments are in some instances overstated. The general context of the study, presented as a re-evaluation of the Warburg hypothesis for NSC-derived tumors, is also somehow misleading for two reasons:

1) Most of the experiments presented are made in the context of normal NSC development.

2) One cannot exclude a more trivial interpretation of the data, whereby OxPhos inhibition triggers a general cellular stress response leading to G_1_/S arrest. In which case, the study fails addressing the role of OxPhos versus glycolytic pathways in the maintenance of stem cell identity or tumor formation, as stated in the title, Abstract and Discussion.

The authors may want to consider rewiring their manuscript and highlight the major strength of the paper, i.e. the demonstration that a proper sequence of cell cycle progression in the NSCs is required for their differentiation program to be completed.

1) The title is misleading. The authors do not address whether OxPhos is sufficient for NSC or tumor proliferation. Therefore, "drive" seems inappropriate. The Abstract is also misleading (see above).

2) Figure 1 lacks an important piece of information: what driver is used for the knockdown? Figure 1I lacks controls RNAi + Complex1 RNAi, which is on the next figure. Inhibition of glycolysis should be presented as a negative control.

3) A control for apoptosis should be presented in the case of tumors.

4) Figure 3: why only 10% NSCs fail to differentiate into Syp-positive, when up to 50% fail to reduce Imp expression upon CI knock-down?

5) Title of the last Results paragraph ("OxPhos dysfunction…) is confusing, as the data is: why do CI or CV silenced-NSCs continue to proliferate in adults if they arrest in G_1_/S during larva? How can one know that the same cells that arrest in G1 in larva are the one that proliferate in adult? More explanation/data is needed to establish this result.

Reviewer #2:

In 2014, Homem et al., reported that in the transition from larval to pupal stages, oxygen consumption rate (OCR) increases and lactate level goes down, and documented a concomitant increase in the expression of genes encoding several enzymes that control rate-limiting steps of related metabolic pathways including the Krebs cycle. They concluded that terminal differentiation of *Drosophila* NSCs is brought about by induction of respiratory chain and oxidative phosphorylation (OxPhos). Moreover, because brat mutant NSCs do not terminally differentiate, Homem and colleagues hypothesised that loss of brat must render NSCs unable to respond to this developmental program.

In the submitted manuscript, van den Ameele and Brand set out to re-asses such a Warburg-like effect both in NSCs and in larval brain tumours. They show that NSCs require OxPhos activity not only for timely termination of cell proliferation, but also, critically, through NSC development to control cell cycle progression, which is essential to control temporal patterning and to ensure neuronal diversity. They also show that OxPhos activity is required to maintain a normal rate of cell proliferation and cell heterogeneity in different types of larval brain tumours. From these data they conclude that both wild-type and tumoral NSCs require OxPhos for proliferation and to generate cellular diversity.

These results challenge those from the previous article on two accounts. Firstly, they very strongly suggest that the key to timely termination of NSC proliferation is not a switch from glycolysis to OxPhos at the onset of pupal life, but substained OxPhos activity throughout NSC development. Secondly they demonstrate three different larval brain tumour types (pros, aPKC-CAAX, and, brat itself) are in fact tightly dependent upon OxPhos. Thus, in short, aerobic glycolysis is not sufficient for normal NSCs development and sustained tumour growth in the *Drosophila* brain.

The reported work has been carried out to high standards; experiments are well designed and controlled; and results are thoroughly documented and conveniently discussed.

These are important results. They are also timely because the Warburg effect it is only too often assumed to be a unescapable trait of human cancer (when in fact it is not) and, by extension of *Drosophila* tumours too.

The one key standing question that the authors have not addressed is whether or not the reported Glycolysis-to-OxPhos transition takes place. The finding that OxPhos is needed throughout NSC development casts serious doubts on the physiological need for such a switch to mostly respiratory metabolism. Moreover, the proposed switch rests on slim evidence, i.e. a 1.2 fold increase in OCR (and that is comparing entire brains, not NSCs).

Other points:

"We assessed whether OxPhos inhibition promotes the differentiation and early depletion of the Imp-positive stem cells. This would result in Imp-negative tumour cells with limited self-renewal capacity (Genovese et al., 2018)". I do not grasp the rationale behind this experiment.

The article "in vivo genetic dissection of tumour growth and the Warburg effect" (Wang et al., 2016) should be cited and discussed. It makes a strong claim on the relevance of the Warburg effect in a different type of tumour in flies.

Figure 1H, At first, I could not appreciate the blue rim on the aPKC-CAAX cell cartoon (it looked all green to me). Using better contrasted colours might help.

Reviewer #3:

This manuscript examines the requirement for OXPHOS during development of *Drosophila* neuronal stem cells. Using an RNAi-based approach, the authors demonstrate that disruption of either complex I or complex V slows both normal brain growth and tumor growth. Moreover, this growth phenotype stems from a decreased cell proliferation and defects in temporal patterning.

Overall, I like the experimental foundation of this paper and I think it will ultimately be quite interesting. While there are experimental shortcomings (see below), the major issue is in regard to the Introduction and Discussion, which fail to place the reader in an appropriate mindset, and as a result, fail to highlight interesting findings. Specifically, the Introduction and Discussion are underdeveloped and devoted a significant number of sentences to providing an overly simplistic discussion of aerobic glycolysis. The text in these sections leads the reader to believe that the paper is devoted to aerobic glycolysis, as evident the final sentence which states that the goal of the study is to examine the role of the Warburg effect in NSCs. Similarly, the first paragraph concludes with a sentence about aerobic glycolysis. But nowhere in the manuscript do the authors examine the role of aerobic glycolysis within NSCs. There are no assays dedicated to studying glycolytic metabolism nor do they disrupt glycolysis in these cells. In fact, they never even measure lactate production – the hallmark byproduct of aerobic glycolysis. I'm not sure why the paper is setup in this manner, other than the possibility that the authors are really annoyed with the inappropriate manner by which other labs sometimes use the term "aerobic glycolysis."

This unwarranted focus on aerobic glycolysis produces a manuscript that lacks direction. All experiments are devoted to studying OXPHOS, yet the reader is focused on glycolysis. I recommend that the authors entirely rewrite the Introduction and Discussion with a focus on the importance of OXPHOS in both cancer and normal developmental growth. OXPHOS is clearly essential in both cancer and developmental growth. In fact, the most recent in vivo 13C-glucose labeling experiments in human patients suggest that most tumors are highly dependent on OXPHOS. Here the authors describe a new system for studying the link between OXPHOS and cell proliferation! Describe why these findings regarding OXPHOS are important and forget about aerobic glycolysis – unless the authors really want to devote a significant amount to experimental effort (far beyond the scope of a simple revision) to prove their hypothesis that aerobic glycolysis (as defined by elevated glucose consumption and oxygen-independent lactate production) is not important in this context.

As a second offshoot of my concern regarding the text, the results herein are clearly related to the studies by Homem et al., 2014, and Mandal et al., 2005, yet, little effort is made to discuss the results in the context of these earlier studies. I don't think that this previous work detracts from the novelty of the study and the lack of discussion is annoying.

My other major concern is that the results are descriptive and lack some rather basic metabolic characterizations. Specifically, there is no metabolic characterization of the cells expressing the complex I and complex V RNAi constructs. Since the manuscript argues that these RNAi treatments are disrupting the cell cycle as the result of defects in OXPHOS, the resulting metabolic defects should be characterized. At a minimum, the authors should examine mitochondrial morphology and ATP levels. All of these parameters can be measured using previously published reagents.

Finally, the Results section would be enhanced with a few simple experiments that would make sense in light of previous CoVa (tenured) studies (Mandal et al., 2005), which demonstrate that disruption of Complex IV results in a G_1_/S arrest that is dependent on AMPK and p53. The authors briefly elude to these similarities, but fail to address the model. A few simple genetic experiments, such as determining if RNAi-depletion of AMPK suppresses the OXPHOS-induced defects, would significantly enhance the impact of the study.

Reviewer #4:

The role of metabolism in modulating development and cancer progression is a fascinating and active area of research. As the authors argue, many studies have focused on the role of aerobic fermentation and one carbon metabolism in fueling growth, but relatively little attention has been paid to OxPhos. In this study the authors argue that oxphos is necessary for proliferation, temporal patterning and differentiation of NSCs. This is a good study that exploits the tools of *Drosophila melanogaster* to address an important question.

I am asking the authors to address the points below:

While the data showing that KD of OxPhos affects different aspects of NSCs are strong, the argument that OxPhos is affected is only genetic. Given the strength of the authors' claim and the fact that changing the mRNA levels of metabolic enzymes often results in compensatory or unforeseen effects, the authors should validate that their manipulations affect OxPhos using different assays. This would make the manuscript much stronger, as there would be direct, rather than genetic (inferential), data about OxPhos.

OxPhos is dispensable for neuronal function, but required in glia (Volkenhoff, 2015). Since NSC can give rise to glial precursors, and glia in turn can support NSCs proliferation and differentiation, is it possible that the phenotypes observed are due to an absence or change in glia?

Given that the brain controls growth of the organism, could changes in brain size, especially when the manipulations result in a smaller brain, be because the larvae are smaller? A way to compare brain size to body length or to another independent metric would be beneficial.

The authors use the word "cause" in the Results and Discussion section; however, their experiments show that OxPhos is necessary for proliferation, not that it is sufficient, thus there is no direct evidence that it "causes or drives" temporal patterning, proliferation, and differentiation.

There is no GAL4 control in the figures.

There is no validation of the efficacy of the RNAi transgenes by qPCR.

---

## [Author Response]

[…] There are several serious criticisms that need to be addressed before the paper can be considered.- A better presentation of the problem.- One major concern is that the paper does not really examine glycolysis; it demonstrates that knocking down two essential components of OxPhos compromises normal NSC development as well as larval brain tumor growth. Refocusing the text on OxPhos while discussing the results in the context of Homem et al. and Mandal et al. would significantly improve the manuscript. It should be noted that the manuscript neither confirms nor discards the previously claimed Glycolysis-to-OxPhos transition.

As suggested by the editor and reviewers 1 and 3, the text has now been refocused on the role of OxPhos in NSCs and brain tumours, and we provide more discussion of Homem et al. and Mandal et al.

- The introduction describes the role of OxPhos in proliferating cells. We introduce normal development of the *Drosophila* CNS and present the *Drosophila* literature on metabolism in tumour cells (Eichenlaub et al., 2018; Wang et al., 2016; Wong et al., 2019). We also introduce the papers from the Banerjee lab (Mandal et al., 2010, 2005; Owusu-Ansah et al., 2008).

- We performed knockdown of glycolysis enzymes in NSCs and in tumour cells and included these data in Figure 1, Figure 1—figure supplement 1, 2 and Figure 2—figure supplement 1.

- The Discussion now clearly states that based on our results we cannot confirm nor discard the presence or absence of the metabolic switch described in Homem et al. However, we also explain that our data provide an alternative interpretation of how OxPhos dysfunction affects termination of NSC proliferation.

- The lack of metabolic characterization of mutants. A revised version should examine mitochondria morphology, their membrane potential, and ATP levels.

To further validate the effect of knockdown of complex I (NDUFS1) and complex V (ATPsynα) on mitochondrial metabolism, we performed the following experiments (Figure 1—figure supplement 2):

- We assessed mitochondrial morphology using STED super-resolution microscopy (conventional confocal microscopy did not resolve individual mitochondria in an intact brain). Both complex I and complex V knockdown showed increased fragmentation of mitochondria in NSCs (Figure 1—figure supplement 2D-F).

- We measured ATP concentration in vivo using a previously validated ATP sensor (Tsuyama et al., 2013). Baseline ATP levels were not affected, but after complex V knockdown, NSCs showed a stronger drop in ATP levels upon acute pharmacological inhibition of glycolysis (Figure 1—figure supplement 2G-I), suggesting that this knockdown affects mitochondrial metabolism and results in rewiring cellular metabolism to rely more on glycolysis.

- One reviewer would also like the paper to address the possibility that these are glial effects.

We did knock down the same complex I and complex V subunits in glia using Repo-Gal4 and assessed the effects on NSCs. In addition, we temporally restricted complex I knockdown to a period in development when cortex glia have already been generated. The results are described in the response to reviewer 4.

- Delete the over-interpretations. In particular, the title must be changed to reflect this: "drive" is an overstatement as the manuscript shows it to be "necessary".

The title, Abstract and Discussion have been changed to reflect that OxPhos is necessary rather than sufficient for the phenotype. The new title is: “Neural stem cell temporal patterning and brain tumour growth rely on oxidative phosphorylation”

The reviews below are quite detailed and should help you improve the manuscript for publication.Reviewer #1:[…] The conclusions of the experiments are in some instances overstated. The general context of the study, presented as a re-evaluation of the Warburg hypothesis for NSC-derived tumors, is also somehow misleading for two reasons:1) Most of the experiments presented are made in the context of normal NSC development.

The Introduction has now been refocused:

- We introduce the importance of OxPhos in proliferating cells.

- We introduce the papers that investigate metabolism of NSCs, in vertebrates as well as in *Drosophila*.

- We describe normal development of the *Drosophila* CNS and highlight the *Drosophila* literature on metabolism in tumour cells (Eichenlaub et al., 2018; Wang et al., 2016; Wong et al., 2019).

- The aim of the paper, as stated at the end of the Introduction now focusses on investigating the role of OxPhos in proliferating NSCs and in tumour cells.

2) One cannot exclude a more trivial interpretation of the data, whereby OxPhos inhibition triggers a general cellular stress response leading to G_1_/S arrest. In which case, the study fails addressing the role of OxPhos versus glycolytic pathways in the maintenance of stem cell identity or tumor formation, as stated in the title, Abstract and Discussion.

We have now performed knockdown of glycolysis enzymes in NSCs and in tumour cells and included these data in Figure 1, Figure 1—figure supplement 1, 2 and Figure 2—figure supplement 1. This shows that NSC-specific knockdown of PFK, aldolase and PGK does not affect brain size, but knockdown of PyK results in smaller brains. We also tested aldolase knockdown in Pros tumours and found that it, in contrast to normal NSCs, their growth is significantly decreased.

We agree that the phenotypes we observe are probably mediated through activation of cellular stress pathways, which, among others, lead to G_1_/S delay, similar to what was shown in Owusu-Ansah et al., 2008 where complex I dysfunction in the *Drosophila* eye disc leads to ROS production and activation of the stress-response through the JNK-pathway.

An in-depth analysis of the pathways mediating this response is beyond the scope of our paper but we have now included the following results:

- Figure 1—figure supplement 2G-I: baseline ATP levels are not decreased upon chronic inhibition of complex V in NSCs. However, ATP levels drop more significantly upon acute inhibition of glycolysis when complex V is inhibited than in the control NSCs. This suggests that part of the response is an increased reliance on glycolysis.

- Figure 4—figure supplement 2: *Ampk* mutation does not rescue proliferation or temporal patterning, and actually makes it worse. This might suggest that AMPK plays a compensatory role in the response to OxPhos dysfunction.

The authors may want to consider rewiring their manuscript and highlight the major strength of the paper, i.e. the demonstration that a proper sequence of cell cycle progression in the NSCs is required for their differentiation program to be completed.

The text has been refocused on the role of OxPhos in NSCs and brain tumours. The Introduction now also describes normal development of the *Drosophila* CNS, and the role of temporal patterning in generation of neuronal diversity. In the Discussion, we provide more explanation of the differences with Homem et al., and discuss how our findings may impact on the current understanding of NSC diversity and normal temporal patterning of NSCs.

1) The title is misleading. The authors do not address whether OxPhos is sufficient for NSC or tumor proliferation. Therefore, "drive" seems inappropriate. The Abstract is also misleading (see above).

The title, Abstract and Discussion have been changed to reflect that OxPhos is necessary rather than sufficient for the phenotype. The new title is: “Neural stem cell temporal patterning and brain tumour growth rely on oxidative phosphorylation”

2) Figure 1 lacks an important piece of information: what driver is used for the knockdown? Figure 1I lacks controls RNAi + Complex1 RNAi, which is on the next figure. Inhibition of glycolysis should be presented as a negative control.

- The driver used for the knockdown is Worniu-Gal4. This is now mentioned in the text and figure legend.

- We performed additional experiments and included the data from the control RNAi + Complex I or V RNAi crosses in Figure 1l.

- We knocked down aldolase in Pros tumours and observed a significant decrease in tumour growth. These data are now included in Figure 1—figure supplement 1A-C.

3) A control for apoptosis should be presented in the case of tumors.

TUNEL staining in the Pros-RNAi brains is now presented in Figure 1—figure supplement 1. We observed no obvious increase in apoptosis upon knockdown of complex I or complex V.

4) Figure 3: why only 10% NSCs fail to differentiate into Syp-positive, when up to 50% fail to reduce Imp expression upon CI knock-down?

The transition from Imp- to Syp-expression is gradual (Liu et al., 2015), and many cells that fail to downregulate Imp upon OxPhos knockdown do express Syp anyway. This is illustrated in Author response image 1 (NSCs with complex I RNAi), whereby Worniu>GFP-positive NSCs that remain Imp-positive often are Syp-positive as well (double-positive NSCs are indicated by arrowheads).

5) Title of the last Results paragraph ("OxPhos dysfunction…) is confusing, as the data is: why do CI or CV silenced-NSCs continue to proliferate in adults if they arrest in G_1_/S during larva? How can one know that the same cells that arrest in G1 in larva are the one that proliferate in adult? More explanation/data is needed to establish this result.

The NSCs do not “arrest” in G_1_ upon expression of Dap or Rbf280. The transition is delayed rather than blocked, since some are in mitosis (pH3+) and they continue to generate neuronal (ElaV+) progeny (See for example in Figure 5—figure supplement 1l).

Adult NSCs are identified by expression of Dpn, as well as by GFP under the control of the NSC-specific Worniu-GAL4. The same GAL4-driver is used for overexpression or knockdown during development and is expressed in more than 90% of the NSCs in the VNC. Because there are no Dpn-positive cells in the adult brain that are GFP-negative (data not shown), it seems unlikely that the few NSCs which did not express to the UAS-transgene are the ones that continue to proliferate in the adult.

The title and the wording of the last Results paragraph have been changed to avoid confusion, and we explain better how adult NSCs are identified.

Reviewer #2:[…] The one key standing question that the authors have not addressed is whether or not the reported Glycolysis-to-OxPhos transition takes place. The finding that OxPhos is needed throughout NSC development casts serious doubts on the physiological need for such a switch to mostly respiratory metabolism. Moreover, the proposed switch rests on slim evidence, i.e. a 1.2 fold increase in OCR (and that is comparing entire brains, not NSCs).

We agree with the reviewer that the existence of such a switch seems less likely in view of our results. We comment on this in the Discussion, and explain an alternative, and in our view more plausible, interpretation of the data: OxPhos-dependent proliferation is required for temporal patterning and differentiation at the G_1_/S transition of the cell cycle; this enables NSCs to undergo normal aging and respond to the developmental cues that instruct termination of proliferation.

Other points:"We assessed whether OxPhos inhibition promotes the differentiation and early depletion of the Imp-positive stem cells. This would result in Imp-negative tumour cells with limited self-renewal capacity (Genovese et al., 2018)". I do not grasp the rationale behind this experiment.

Growth of pros mutant tumours is known to be sustained by a small proportion of highly proliferative stem cells expressing Imp (Genovese et al., 2018; Narbonne-Reveau et al., 2016). These Imp-positive cells self-renew, but also generate more differentiated tumour-cells that limit tumour growth. Because OxPhos dysfunction reduces tumour growth (Figure 1A-J), we wanted to exclude the possibility that this was a result of increased differentiation towards Imp-negative cells with limited self-renewal capacity. This is now explained in the Results section.

The article "in vivo genetic dissection of tumour growth and the Warburg effect" (Wang et al., 2016) should be cited and discussed. It makes a strong claim on the relevance of the Warburg effect in a different type of tumour in flies.

The article is now presented in the Introduction.

Figure 1H, At first, I could not appreciate the blue rim on the aPKC-CAAX cell cartoon (it looked all green to me). Using better contrasted colours might help.

The colour in the figure has been changed.

Reviewer #3:This manuscript examines the requirement for OXPHOS during development of Drosophila neuronal stem cells. Using an RNAi-based approach, the authors demonstrate that disruption of either complex I or complex V slows both normal brain growth and tumor growth. Moreover, this growth phenotype stems from a decreased cell proliferation and defects in temporal patterning.Overall, I like the experimental foundation of this paper and I think it will ultimately be quite interesting. While there are experimental shortcomings (see below), the major issue is in regard to the Introduction and Discussion, which fail to place the reader in an appropriate mindset, and as a result, fail to highlight interesting findings. Specifically, the Introduction and Discussion are underdeveloped and devoted a significant number of sentences to providing an overly simplistic discussion of aerobic glycolysis. The text in these sections leads the reader to believe that the paper is devoted to aerobic glycolysis, as evident the final sentence which states that the goal of the study is to examine the role of the Warburg effect in NSCs. Similarly, the first paragraph concludes with a sentence about aerobic glycolysis. But nowhere in the manuscript do the authors examine the role of aerobic glycolysis within NSCs. There are no assays dedicated to studying glycolytic metabolism nor do they disrupt glycolysis in these cells. In fact, they never even measure lactate production – the hallmark byproduct of aerobic glycolysis. I'm not sure why the paper is setup in this manner, other than the possibility that the authors are really annoyed with the inappropriate manner by which other labs sometimes use the term "aerobic glycolysis."This unwarranted focus on aerobic glycolysis produces a manuscript that lacks direction. All experiments are devoted to studying OXPHOS, yet the reader is focused on glycolysis. I recommend that the authors entirely rewrite the Introduction and Discussion with a focus on the importance of OXPHOS in both cancer and normal developmental growth. OXPHOS is clearly essential in both cancer and developmental growth. In fact, the most recent in vivo 13C-glucose labeling experiments in human patients suggest that most tumors are highly dependent on OXPHOS. Here the authors describe a new system for studying the link between OXPHOS and cell proliferation! Describe why these findings regarding OxPhos are important and forget about aerobic glycolysis – unless the authors really want to devote a significant amount to experimental effort (far beyond the scope of a simple revision) to prove their hypothesis that aerobic glycolysis (as defined by elevated glucose consumption and oxygen-independent lactate production) is not important in this context.

We have refocused the manuscript on the importance of OxPhos in proliferating NSCs.

- The Introduction explains the importance of OxPhos in proliferating cells.

- The Discussion has been expanded considerably. We provide more explanation of the discrepancies with Homem et al. We discuss how our findings may impact on the current understanding of NSC diversity and normal temporal patterning of NSCs. We also present our findings in the context of research from the Banerjee lab (Mandal et al., 2010, 2005; Owusu-Ansah et al., 2008).

- We have now performed knockdown of glycolysis enzymes in NSCs and in tumour cells and included these data in Figure 1, Figure 1—figure supplement 1, 2 and Figure 2—figure supplement 1. This shows that NSC-specific knockdown of PFK, aldolase and PGK does not affect brain size, but knockdown of PyK results in smaller brains. We also tested aldolase knockdown in Pros tumours and found that, in contrast to normal NSCs, their growth is significantly decreased.

As a second offshoot of my concern regarding the text, the results herein are clearly related to the studies by Homem et al., 2014, and Mandal et al., 2005, yet, little effort is made to discuss the results in the context of these earlier studies. I don't think that this previous work detracts from the novelty of the study and the lack of discussion is annoying.

- Mandal et al. and Homem et al. are better explained in the Introduction, and we mention both papers as a motivation to conduct our experiments.

- We highlight the differences with Homem et al. and discuss in more detail an alternative interpretation of their data based on our findings.

- Additional experiments (cf below), based on the work from Mandal et al., have now been included in the Results section (Figure 4—figure supplement 2) and are interpreted in the context of (Mandal et al., 2010, 2005; Owusu-Ansah et al., 2008).

My other major concern is that the results are descriptive and lack some rather basic metabolic characterizations. Specifically, there is no metabolic characterization of the cells expressing the complex I and complex V RNAi constructs. Since the manuscript argues that these RNAi treatments are disrupting the cell cycle as the result of defects in OXPHOS, the resulting metabolic defects should be characterized. At a minimum, the authors should examine mitochondrial morphology and ATP levels. All of these parameters can be measured using previously published reagents.

The complex I RNAi line has been used extensively and validated in several previous publications (e.g. Garcia et al., 2017; Hermle et al., 2017; Owusu-Ansah et al., 2013; Pletcher et al., 2019), which we now cite in the Results section.

To further validate the effect of knockdown of complex I (NDUFS1) and complex V (ATPsynα) on mitochondrial metabolism, we performed the following experiments (Figure 1—figure supplement 2):

- We assessed mitochondrial morphology using STED super-resolution microscopy (conventional confocal microscopy did not resolve individual mitochondria in an intact brain). Both complex I and complex V knockdown showed increased fragmentation of mitochondria in NSCs (Figure 1—figure supplement 2D-F).

- We measured ATP concentration in vivo using a previously validated ATP sensor (Tsuyama et al., 2013). Baseline ATP levels were not affected, but after complex V knockdown, NSCs showed a stronger drop in ATP levels upon acute pharmacological inhibition of glycolysis (Figure 1—figure supplement 2G-I), suggesting that this knockdown affects mitochondrial metabolism and results in rewiring cellular metabolism to rely more on glycolysis.

Finally, the Results section would be enhanced with a few simple experiments that would make sense in light of previous CoVa (tenured) studies (Mandal et al., 2005), which demonstrate that disruption of Complex IV results in a G_1_/S arrest that is dependent on AMPK and p53. The authors briefly elude to these similarities, but fail to address the model. A few simple genetic experiments, such as determining if RNAi-depletion of AMPK suppresses the OXPHOS-induced defects, would significantly enhance the impact of the study.

In view of the reviewers’ comments, we assessed both retrograde signalling pathways that were shown to cause G_1_/S arrest by (Mandal et al., 2010, 2005; Owusu-Ansah et al., 2008). Our preliminary data suggest that decreasing ROS does not rescue the proliferation or temporal patterning defects of complex I or V inhibition (data not shown). We also knocked down Ampk and p53 by RNAi, and these did not rescue the effect of OxPhos knockdown (complex I RNAi) either (data not shown).

We next generated Ampk mutant clones through mitotic recombination, in a background where all NSCs continue to express the complex I or complex V RNAi. This again did not rescue proliferation or temporal patterning, and in fact enhanced rather than suppressed the phenotype. We included these experiments in the Results section (Figure 4—figure supplement 2) and interpret them in the context of (Mandal et al., 2010, 2005; Owusu-Ansah et al., 2008).

Reviewer #4:[…] I am asking the authors to address the points below:While the data showing that KD of OxPhos affects different aspects of NSCs are strong, the argument that OxPhos is affected is only genetic. Given the strength of the authors' claim and the fact that changing the mRNA levels of metabolic enzymes often results in compensatory or unforeseen effects, the authors should validate that their manipulations affect OxPhos using different assays. This would make the manuscript much stronger, as there would be direct, rather than genetic (inferential), data about OxPhos.

The complex I RNAi line has been used extensively and validated in several previous publications (e.g. Garcia et al., 2017; Hermle et al., 2017; Owusu-Ansah et al., 2013; Pletcher et al., 2019), which we now cite in the Results section.

To further validate the effect of knockdown of complex I (NDUFS1) and complex V (ATPsynα) on mitochondrial metabolism, we performed the following experiments (Figure 1—figure supplement 2):

- We assessed mitochondrial morphology using STED super-resolution microscopy (conventional confocal microscopy did not resolve individual mitochondria in an intact brain). Both complex I and complex V knockdown showed increased fragmentation of mitochondria in NSCs (Figure 1—figure supplement 2D-F).

- We measured ATP concentration in vivo using a previously validated ATP sensor (Tsuyama et al., 2013). Baseline ATP levels were not affected, but after complex V knockdown, NSCs showed a stronger drop in ATP levels upon acute pharmacological inhibition of glycolysis (Figure 1—figure supplement 2G-I), suggesting that this knockdown affects mitochondrial metabolism and results in rewiring cellular metabolism to rely more on glycolysis.

OxPhos is dispensable for neuronal function, but required in glia (Volkenhoff, 2015). Since NSC can give rise to glial precursors, and glia in turn can support NSCs proliferation and differentiation, is it possible that the phenotypes observed are due to an absence or change in glia?

All literature so far, in mammals and *Drosophila* (recently reviewed in Magistretti and Alleman, 2018), including the paper mentioned (Volkenhoff et al., 2015), suggest that glial metabolism is mainly glycolytic, to support the function of neurons through secretion of lactate and alanine.

Based on the reviewer’s suggestion, we knocked down the same complex I and complex V subunits in glia using Repo-Gal4. Glial knockdown resulted in a decreased mitotic index of NSCs (Author response image 2). However this did not affect temporal patterning, as assessed by the number of Imp-positive NSCs at the end of neurogenesis (Author response image 2).

In addition, we temporally restricted complex I knockdown, and expressed the RNAi from L1 onwards. At this stage, all cortex glia have already been generated and thus are not affected by the expression of RNAi in NSCs. This also prevented termination of proliferation as shown in Author response image 3 and indicates that the phenotype is not due to an effect on glia.

**Author response image 2. respfig2:** 

**Author response image 3. respfig3:** 

Given that the brain controls growth of the organism, could changes in brain size, especially when the manipulations result in a smaller brain, be because the larvae are smaller? A way to compare brain size to body length or to another independent metric would be beneficial.

L3 larvae were always collected at wandering stage in order to synchronize developmental timing. In addition, we quantified larval length at the stage of dissection, and pupal length as a measure for final developmental size. The results are included in Figure 2—figure supplement 2 and did not reveal significant differences between the experimental conditions.

The authors use the word "cause" in the Results and Discussion section; however, their experiments show that OxPhos is necessary for proliferation, not that it is sufficient, thus there is no direct evidence that it "causes or drives" temporal patterning, proliferation, and differentiation.

The title, Abstract and Discussion have been changed to reflect that OxPhos is necessary rather than sufficient for the phenotype. The new title is: “Neural stem cell temporal patterning and brain tumour growth rely on oxidative phosphorylation”

There is no GAL4 control in the figures.

As a control, we crossed the same GAL4-line to an RNAi against mCherry. This controls for the presence of GAL4, the RNAi and the genetic background.

There is no validation of the efficacy of the RNAi transgenes by qPCR.

- The complex I RNAi line has been extensively used and validated in several previous publications (e.g. Garcia et al., 2017; Hermle et al., 2017; Owusu-Ansah et al., 2013; Pletcher et al., 2019), which we now mention in the Results section.

- We now validated the complex V RNAi line (against Blw) by staining for the Blw protein (ATPsynα), which showed a clear downregulation (Figure 1—figure supplement 2A-C).

- We also performed a more functional validation of the RNAi-lines as explained above.